# The state of iodine deficiency in Kyrgyzstan: Insights from studies of knowledge, attitudes and practices

Yulia Chyngyshpaeva[1], Zhokhongir Dzhaliev[1], Zhyparkul Derbishalieva[1], Muiz Ibrahim[1], Don Eliseo III Lucero-Prisno[2,3,4], Kenesh Dzhusupov[1,5]*

1 Department of Public Health, International Higher School of Medicine, Bishkek, Kyrgyzstan, 2 Department of Global Health and Development, Faculty of Public Health and Policy, London School of Hygiene and Tropical Medicine, London, United Kingdom, 3 Office for Research, Innovation and Extension Services, Southern Leyte State University, Sogod, Southern Leyte, Philippines, 4 Center for University Research, University of Makati, Makati City, Philippines, 5 Department of Public Health, Osh State University, Osh, Kyrgyzstan

* k.dzhusupov@ism.edu.kg

**Data Availability Statement:** "All relevant data are within the paper and its Supporting information files".

## Abstract

### Objective

This study aimed to assess the knowledge, attitudes, and practices (KAPs) regarding iodine deficiency among the population of Kyrgyzstan, with a focus on pregnant women, and to correlate these factors with the urinary iodine concentration (UIC).

### Methods

A cross-sectional study was conducted using structured questionnaires and urine sample analysis to evaluate iodine status. A multistage stratified sampling method ensured a representative sample from both urban and rural areas. Descriptive statistics were used to summarize demographic characteristics and KAP variables, whereas chi-square tests and multivariate logistic regression analysis were used to identify predictors of KAP outcomes.

### Results

The study included 690 participants, with balanced representation from both urban and rural areas. The mean knowledge score was moderate, with significant gaps in the understanding of iodine deficiency. The participants generally had positive attitudes toward iodine intake but exhibited suboptimal practices, especially in rural areas. Pregnant women demonstrated greater knowledge, but fewer practical behaviors related to iodine intake. Compared with rural participants, urban participants had higher UIC levels, indicating better iodine nutrition. Multivariate regression analysis revealed that residence, knowledge, attitudes, and practices significantly influenced UIC levels.

### Conclusion

Despite existing iodization programs, significant gaps remain in the knowledge and practices related to iodine intake, particularly in rural areas. This study highlights the need for

**Funding:** The author(s) received no specific funding for this work.

**Competing interests:** The authors have declared that no competing interests exist.

targeted public health interventions to improve iodine nutrition and reduce the burden of iodine deficiency disorders in Kyrgyzstan. Enhancing public health education, improving the accessibility and affordability of iodized salt, and regularly monitoring the iodine status are essential strategies for addressing these issues.

## Introduction

Iodine is an essential micronutrient required for the synthesis of thyroid hormones, which are crucial for normal growth, development, and metabolic regulation [1]. Iodine deficiency remains a significant public health issue worldwide, particularly in regions with insufficient natural iodine sources.

Iodine deficiency can lead to a spectrum of disorders collectively known as iodine deficiency disorders (IDDs), including goiter, hypothyroidism, and various cognitive impairments. The most vulnerable groups are pregnant women, infants, and children, who require sufficient iodine for neurodevelopment and growth [1].

Globally, iodine deficiency remains a concern despite efforts such as salt iodization programs, which have significantly reduced the prevalence of IDDs over the past decades. However, recent studies indicate a resurgence of iodine deficiency in in certain populations in some developed countries, suggesting that ongoing vigilance and public health interventions are necessary to maintain adequate iodine levels, especially in rural areas [2–6]. This re-emergence underscores the importance of assessing and addressing iodine deficiency in regions characterized by biogeoendemic conditions, such as Kyrgyzstan, where IDDs remain a persistent issue. In Kyrgyzstan, efforts to combat iodine deficiency have included mandatory iodization of table salt, which has shown some success in reducing the prevalence of IDDs. However, this problem persists, indicating the need for ongoing monitoring and interventions. A study conducted by Orozbekov et al. (2021) highlighted that despite these efforts, a significant portion of the population, particularly in rural areas, continues to suffer from iodine deficiency due to limited awareness and access to iodized salt [7].

Cultural factors, such as dietary habits and access to iodized salt, can differ significantly between urban and rural populations in Kyrgyzstan. Cultural beliefs and practices around food, particularly in rural areas, may limit the use of iodized salt, exacerbating iodine deficiency. This study addresses these factors by analyzing knowledge, attitudes, and practices (KAP) across diverse geographic and cultural contexts in Kyrgyzstan [1, 2].

Understanding the knowledge, attitudes, and practices (KAPs) of populations regarding iodine deficiency is vital for designing effective public health interventions. KAP studies can help identify gaps in awareness and behavior that can be targeted through education and policy measures. For example, a comprehensive review highlighted that many women of childbearing age lack knowledge about the importance of iodine and do not consistently receive information from healthcare professionals [8]. Such insights are critical for tailoring public health strategies to improve iodine intake and reduce the burden of IDDs.

This study aimed to provide an in-depth analysis of the current state of knowledge, attitudes, and practices (KAP) regarding iodine deficiency among the Kyrgyzstani population. By addressing this aim, this study sought to contribute to the body of knowledge on iodine deficiency and inform public health strategies to mitigate the risks associated with inadequate iodine intake in Kyrgyzstan.

## Participants and methods

### Study design

This cross-sectional study was designed to assess the knowledge, attitudes, and practices (KAPs) regarding iodine deficiency among the population of Kyrgyzstan, with a focus on pregnant women due to their heightened vulnerability to iodine deficiency disorders (IDDs). The study utilized structured questionnaires and urine sample collection to evaluate iodine status and correlate it with KAP outcomes.

### Study site

This study was conducted between June 26, 2021, and April 18, 2022, across various regions of Kyrgyzstan, encompassing both urban and rural areas, to ensure a comprehensive representation of the population. The urban areas included major cities (Bishkek and Osh), whereas the rural areas were selected from different areas to reflect diverse socioeconomic and geographic characteristics.

### Sampling method

A multistage stratified sampling method was employed to ensure that the sample was representative of the population in terms of age, sex, and geographical distribution. The study population was stratified into urban and rural areas. Random clusters were selected within each stratum, with a greater number of clusters from urban areas owing to their higher population density. Within each cluster, households were randomly selected. One eligible participant from each household was selected via a random selection method.

The sample size for a cross-sectional study was calculated via the following formula:

$$n = Z^2 \cdot P \cdot \frac{(1 - P)}{d^2}$$

where:

- **n** is the sample size.

- **Z** is the Z value (1.96 for the 95% confidence level).

- **P** is the estimated prevalence of adequate iodine knowledge (assumed to be 50% owing to the lack of precise estimates).

- **d** is the margin of error (5%).

$$n = 1.96^2 \cdot 0.5 \cdot \frac{(1 - 0.5)}{0.05^2} = 384$$

Considering a design effect of 1.5 to account for the complex sampling design and nonresponse rate of 10%, the final sample size was adjusted as follows:

$$n_{adjusted} = 384 \cdot 1.5 \cdot \frac{100}{90} = 640$$

To ensure representation from both urban and rural populations, the sample was further stratified on the basis of the proportion of urban (2.34 million) and rural (4.116 million) populations:

- Urban sample: 2.34/6.456 * 640 = 232

- Rural sample: 4.116/6.456 * 640 = 408

    From the 726 respondents, we selected 690 with full answers to the questionnaire.

## Inclusion and exclusion criteria

We included in the study:

- Residents of Kyrgyzstan aged 18 years and above.

- Preference was given to pregnant women.

- Individuals who have lived in the selected area for at least one year.

- Willingness to provide a urine sample for iodine analysis.

    We excluded individuals with cognitive impairments that hinder their ability to understand and respond to the questionnaire and nonresidents or those living in the area for less than one year.

## Data collection instruments and procedure

To collect data, we used the semistructured Food and Agriculture Organization (FAO) of the United Nations nutrition-related iodine deficiency questionnaire adapted for the specific objectives of the study [9]. The questionnaire was designed to capture demographic information, knowledge about iodine deficiency, attitudes toward iodine intake, practices related to iodine consumption, and health status. The questionnaire was pretested in a pilot study to ensure clarity and reliability. The pilot study led to significant modifications in the final questionnaire, particularly in the phrasing of questions related to dietary practices. The results highlighted the need to simplify complex medical terms for participants with lower educational levels, leading to adjustments in the language used throughout the questionnaire.

    The data collectors received comprehensive training on the study objectives, ethical considerations, and interview techniques to ensure consistency and accuracy. The questionnaire was validated through expert review and a pilot study, ensuring both clarity and reliability. Trained data collectors conducted face–to–face interviews via structured questionnaires. Urine samples were collected from the participants to assess their iodine status.

## Urine sampling and analysis

Clean, midstream urine samples were collected from each participant in sterile containers, stored in a cool environment (4˚C) and transported to the laboratory within 24 hours of collection. The urinary iodine concentration (UIC) was measured by the Hemotest laboratory via the Sandell–Kolthoff reaction, a spectrophotometric method widely used for iodine determination in urine. This method involves the catalytic reduction of ceric ammonium sulfate in the presence of arsenious acid and is sensitive to variations in the iodine concentration. A UIC <100 μg/L is considered insufficient for adults, and a UIC <150 μg/L is considered insufficient for pregnant females.

## Statistical analysis methods

The collected data were analyzed via the IBM Statistical Package for the Social Sciences version 29.0. Descriptive statistics were used to summarize demographic characteristics and KAP variables. Chi-square tests were used to assess the associations between demographic factors and KAP outcomes. Statistical significance was set at P <0.05. The rationale for using multivariate regression was to account for potential confounders, ensuring robust analysis of predictors of

urinary iodine concentration (UIC), such as residence, education, and socioeconomic status (Patil et al., 2019). Multivariate logistic regression analysis was performed to identify the independent predictors of good knowledge, positive attitudes, and appropriate practices related to iodine intake. We chose chi-square tests and multivariate logistic regression due to their suitability in analyzing categorical and continuous variables, respectively, ensuring robust identification of predictors and reliability of the data. Multicollinearity between predictors was assessed using variance inflation factors (VIF). All VIF values were below 2, indicating no severe multicollinearity issues. This ensured the reliability of the regression coefficients, particularly when evaluating the significance of individual predictors of UIC (Lisco et al., 2023).

The knowledge score was calculated on the basis of participants' responses to questions designed to assess their knowledge about iodine deficiency, its sources, and health implications. The attitude score was derived from participants' responses to questions reflecting their beliefs and feelings about iodine intake and its importance. The practice score was based on participants' responses to questions about their behaviors and practices related to iodine intake. Each correct answer on knowledge and practice was assigned a score of 1, and incorrect or "do not know" answers received a score of 0. The total score was the sum of all correct answers. The questions on attitudes used a Likert scale (e.g., 1 = Strongly Disagree, 2 = Disagree, 3 = Neutral, 4 = Agree, and 5 = Strongly Agree). The scores for each item were summed to produce the total attitude scores. The assumptions of normality for the KAP scores and UIC distribution were examined using the Shapiro-Wilk test. Both variables followed a normal distribution, validating their treatment as continuous variables in the regression analysis [13].

This study provides a framework for calculating and interpreting KAP scores in health research, ensuring that the scoring system is both reliable and valid.

## Ethical considerations

The study was conducted in accordance with the ethical principles outlined in the Declaration of Helsinki. Ethical approval was obtained from the Ethics Review Committee (protocol REC-3, 21.01.2020) of the International Higher School of Medicine (Bishkek, Kyrgyzstan). Written informed consent was obtained from all participants prior to the interviews. The participants were assured of the confidentiality and anonymity of their responses. They were informed that their participation was voluntary and that they could withdraw from the study at any time without any consequences.

By adhering to these methodological standards, this study aimed to provide reliable and valid data on KAP regarding iodine deficiency among the population, thereby informing public health strategies to improve iodine intake and reduce the burden of IDDs in Kyrgyzstan.

## Results

### Sociodemographic characteristics of the respondents

The study included 690 participants, with a balanced representation of urban and rural areas. The majority were aged between 26 and 35 years, and females outnumbered males slightly. The educational level varied, with a significant portion having a higher education level.

The sociodemographic characteristics of the studied population are presented in Table 1.

Table 2 provides a detailed breakdown of the characteristics of the pregnant female population in urban and rural areas, including age distribution, sex, education level, ethnicity, monthly family income per capita, and occupation status.

**Table 1. Socio-demographic characteristics of the study population.**

| Characteristic | Urban (%) | Rural (%) | Total (%) |
|---|---|---|---|
| **Area** | 327 (47.39) | 140 (51.28) | 690 (100) |
| **Age** | | | |
| 18–25 | 76 (23.24) | 78 (21.49) | 154 (22.32) |
| 26–35 | 148 (45.26) | 151 (41.6) | 299 (43.33) |
| 36–45 | 85 (25.99) | 90 (24.79) | 175 (25.36) |
| 46–55 | 12 (3.67) | 29 (7.99) | 41 (5.94) |
| 56 and above | 6 (1.83) | 15 (4.13) | 21 (3.04) |
| **Gender** | | | |
| Male | 149 (45.57) | 182 (50.14) | 321 (47.97) |
| Female | 178 (54.43) | 181 (49.86) | 359 (52.03) |
| incl.pregnant | 133 | 140 | 273 |
| **Education Level** | | | |
| Primary education | 2 (0.61) | 4 (1.1) | 6 (0.87) |
| Secondary school education | 4 (1.22) | 19 (5.23) | 23 (3.33) |
| High school education | 225 (68.81) | 248 (68.32) | 473 (68.55) |
| Higher education | 96 (29.36) | 92 (25.34) | 188 (27.25) |
| **Ethnicity** | | | |
| Kyrgyzs | 219 (66.97) | 308 (84.85) | 527 (76.38) |
| Uzbeks | 45 (13.76) | 26 (7.16) | 71 (10.29) |
| Russians | 36 (11.01) | 7 (1.93) | 43 (6.23) |
| Dungans | 2 (0.61) | 14 (3.86) | 16 (2.32) |
| Others | 25 (7.65) | 8 (2.2) | 33 (0.58) |
| **Monthly family income per capita KGS** | | | |
| <8000 | 89 (27.22) | 139 (38.29) | 228 (33.04) |
| 8001–16000 | 102 (31.19) | 165 (45.45) | 267 (38.7) |
| 16001–30000 | 85 (25.99) | 54 (14.88) | 139 (20.14) |
| >30000 | 51 (15.60) | 4 (1.1) | 55 (7.97) |
| **Occupation status** | | | |
| Student | 67 (20.49) | 63 (17.36) | 130 (18.84) |
| Employed | 188 (57.49) | 141 (38.84) | 329 (47.68) |
| Unemployed | 68 (20.80) | 146 (40.22) | 214 (31.02) |
| Retired | 4 (1.22) | 13 (3.58) | 17 (2.46) |

## KAP scores

The mean KAP scores related to iodine are presented in Table 3. The mean knowledge score was moderate, with significant gaps identified in the understanding of iodine deficiency and its sources. The knowledge scores were moderate, and while attitudes were generally positive, there was a noted discrepancy between knowledge and actual practices.

The participants generally had positive attitudes toward iodine intake, recognizing its importance for health. However, the practice scores were suboptimal, indicating inadequate behaviors related to iodine intake, particularly in rural areas.

As shown in Table 3, pregnant women in rural areas seem to have marginally better knowledge about iodine-related health issues than their urban counterparts do. Pregnant females in urban areas have more positive attitudes toward iodine-related health issues than do those in rural areas. Both urban and rural pregnant females have similar practical behaviors related to iodine consumption, reflecting consistent practices across these areas.

**Table 2. Demographic characteristics of the pregnant women in the study population.**

| Characteristic | Urban (%) | Rural (%) | Total (%) |
|---|---|---|---|
| Area | 133 (48.72) | 140 (52.61) | 273 (100) |
| **Age** | | | |
| 18–25 | 38 (28.57) | 38 (27.14) | 76 (27.84) |
| 26–35 | 68 (51.13) | 59 (42.14) | 127 (46.52) |
| **36–45** | 27 (20.3) | 43 (31.72) | 70 (25.64) |
| **Education Level** | | | |
| Primary education | 2 (1.50) | 4 (2.86) | 6 (2.20) |
| Secondary school education | 4 (3.01) | 19 (13.57) | 23 (8.42) |
| High school education | 96 (72.18) | 92 (65.71) | 188 (68.86) |
| Higher education | 31 (23.31) | 25 (17.86) | 56 (20.52) |
| **Ethnicity** | | | |
| Kyrgyzs | 97 (72.93) | 104 (74.29) | 201 (73.63) |
| Uzbeks | 10 (7.52) | 15 (10.71) | 25 (9.16) |
| Russians | 14 (10.53) | 10 (7.14) | 24 (8.79) |
| Dungans | 2 (1.50) | 5 (3.57) | 7 (2.56) |
| Others | 10 (7.52) | 6 (4.29) | 16 (5.86) |
| **Monthly family income per capita KGS** | | | |
| <8000 | 27 (20.30) | 60 (42.86) | 87 (31.87) |
| 8001–16000 | 35 (26.32) | 49 (35.00) | 84 (30.77) |
| 16001–30000 | 31 (23.30) | 20 (14.29) | 51 (18.68) |
| >30000 | 40 (30.08) | 11 (7.86) | 51 (18.68) |
| **Occupation status** | | | |
| Student | 23 (17.29) | 15 (10.71) | 38 (13.92) |
| Employed | 79 (59.40) | 59 (42.14) | 138 (50.55) |
| Unemployed | 31 (23.31) | 66 (47.15) | 97 (34.53) |

The KAP scores of the different sociodemographic groups are shown in Figs 1 and 2.

## Knowledge related to iodine deficiency

As shown in Fig 1, knowledge scores vary across different age groups. The younger age groups (1 and 2) tended to have slightly lower median scores than the older age groups did. A positive correlation with age indicates that knowledge scores tend to increase with age. Both males and females have similar median knowledge scores, with slightly wider variability in females. Compared with rural residents, urban residents have slightly higher median knowledge scores.

**Table 3. KAP scores of the respondents, including pregnant women in Kyrgyzstan, regarding iodine deficiency.**

| Variable | Respondents | Urban (Mean ± SD) | Rural (Mean ± SD) | Total (Mean ± SD) |
|---|---|---|---|---|
| **Knowledge Score** | All participants | 2.61 ± 0.92 | 2.74 ± 0.83 | 2.68 ± 0.88 |
| | Pregnant women | 3.12 ± 0.72 | 2.85 ± 0.65 | 2.98 ± 0.69 |
| **Attitude Score** | All participants | 3.16 ± 0.96 | 2.83 ± 0.98 | 2.99 ± 0.98 |
| | Pregnant women | 3.75 ± 0.85 | 3.45 ± 0.78 | 3.60 ± 0.82 |
| **Practice Score** | All participants | 3.73 ± 0.73 | 3.73 ± 0.66 | 3.73 ± 0.72 |
| | Pregnant women | 3.10 ± 0.50 | 2.75 ± 0.55 | 2.93 ± 0.53 |

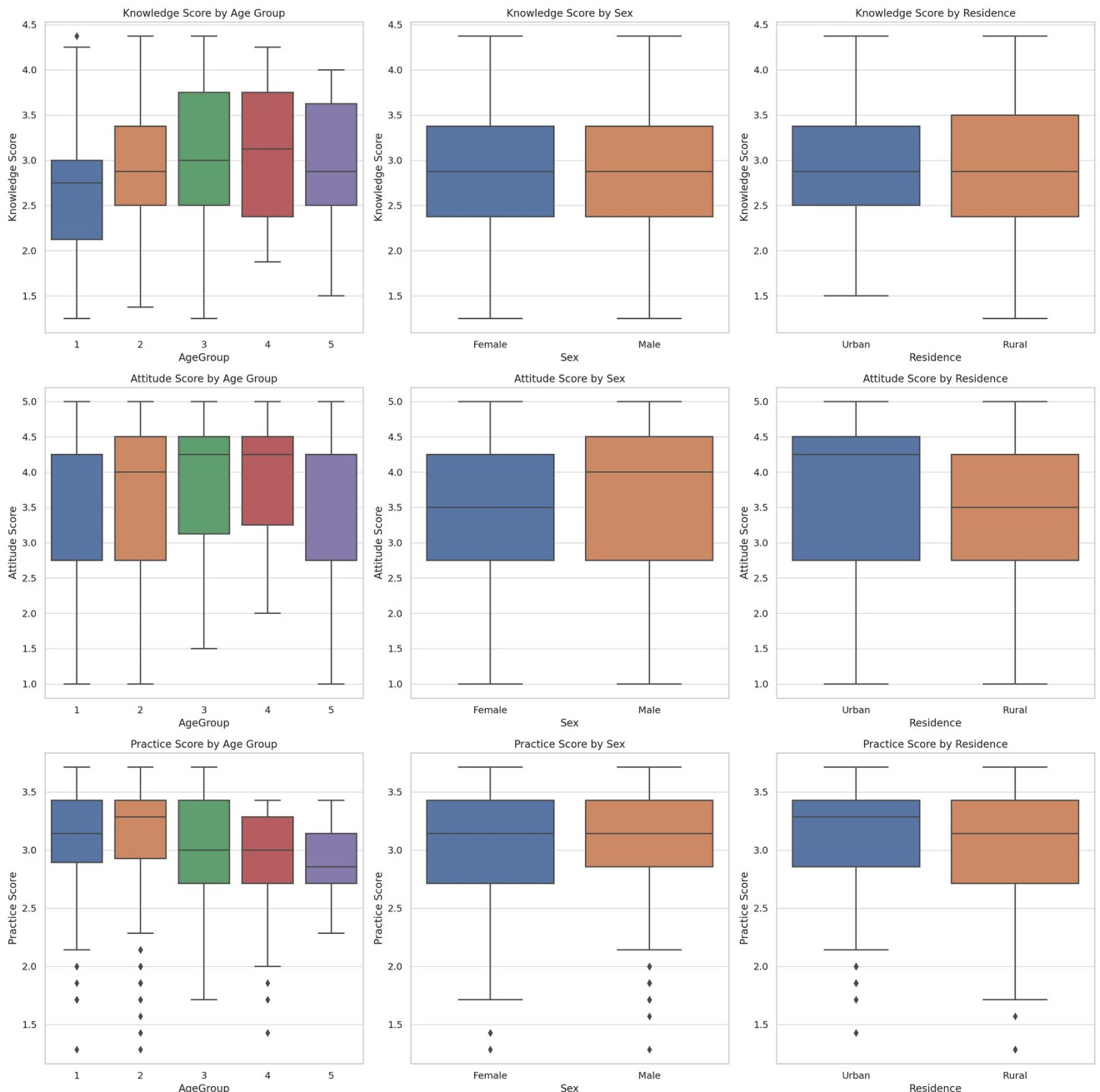

**Fig 1. KAP scores of respondents in Kyrgyzstan regarding iodine deficiency by age, sex and residency group.**

In Fig 2, the KAP scores depending on ethnicity, income and education level are presented. Ethnic groups such as the Russian, Kyrgyz, and Uzbek groups have similar median knowledge scores, whereas the 'Others' and Dungan groups have slightly lower scores. Respondents in higher income groups (upper-middle income and high income) have higher median knowledge scores than do those in lower income groups. The participants with higher education levels had significantly higher knowledge scores, whereas the primary and high school education groups had lower scores.

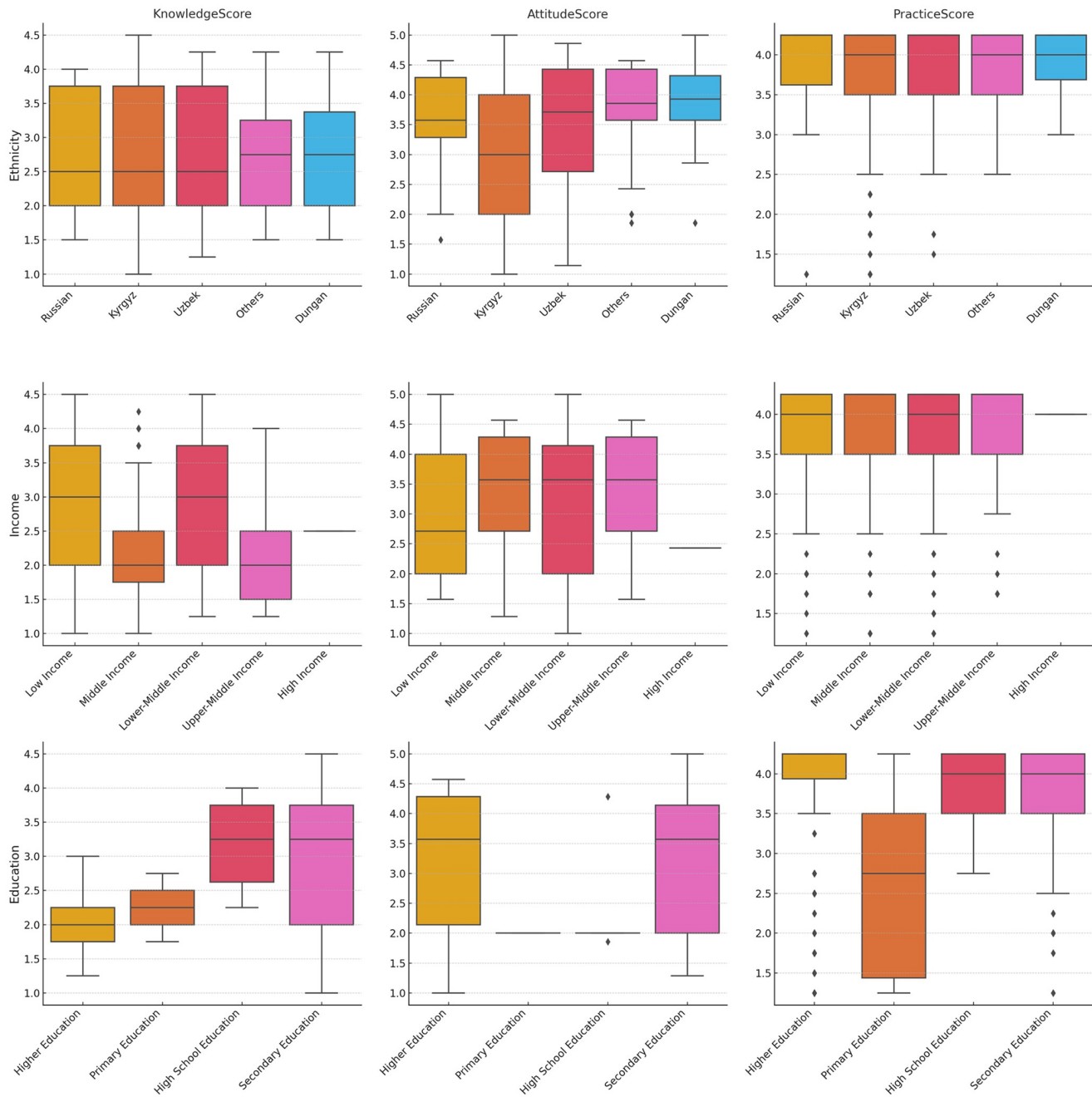

**Fig 2. KAP scores of respondents in Kyrgyzstan regarding iodine deficiency by ethnicity, income and education group.**

### Attitudes related to iodine deficiency

The attitude scores are relatively consistent across age groups, with a slight increase in the older age groups. Both males and females have similar median attitude scores, with males showing slightly greater variability. Compared with rural residents, urban residents have higher median attitude scores (Fig 1).

As presented in Fig 2, the attitude scores are higher for the Kyrgyz and Uzbek groups, indicating more positive attitudes toward iodine. Moreover, attitude scores also increase with

income, with higher income groups showing more positive attitudes. Attitudes toward iodine improve with higher education levels, with higher education groups showing the most positive attitudes.

## Practice related to iodine deficiency

Practice scores vary across age groups, with younger age groups showing more variability. Both males and females have similar practice scores, with slightly wider variability in males. Compared with rural residents, urban residents have slightly higher median practice scores (Fig 1).

Practice scores are relatively consistent across ethnic groups, with the Kyrgyz and Uzbek groups showing slightly higher median scores (Fig 2). Higher income groups have better practices related to iodine consumption, as indicated by higher median scores. Practice scores are higher for individuals with higher education levels, indicating better iodine-related practices.

In Fig 3, a correlation matrix heatmap represents the correlation coefficients between the sociodemographic characteristics and the KAP scores of the respondents. These coefficients indicate the strength and direction of a linear relationship between two variables.

The study results suggest that individuals with higher knowledge scores tend to have higher attitude scores. Better attitudes correlate with better practices. Higher knowledge scores are somewhat associated with better practice scores.

There was no significant difference in the KAP scores based on sex. Older individuals might have slightly higher knowledge and attitude scores. However, younger individuals might have slightly better practice scores. Urban individuals might have slightly better scores for knowledge, attitudes, and practices.

Specific ethnic groups (Uzbeks, Russians, Tajiks) tend to have higher knowledge and attitude scores, highlighting cultural or community influences. Higher income is associated with better attitudes and practices.

## Urinary iodine concentrations

Table 4 presents the urinary iodine concentration (UIC) for the study population (numerator) and pregnant females (denominator), segmented by urban and rural residence, along with the percentages of sufficient and insufficient UIC levels. The mean UIC was within the insufficient range for a significant portion of the study population, particularly in rural areas, indicating ongoing iodine deficiency despite mandatory iodization programs.

Compared with rural females, urban females have a greater average UIC, indicating better iodine nutrition among urban residents. Compared with rural females, a significantly greater percentage of urban females have sufficient iodine levels, suggesting disparities in iodine nutrition based on residence. Compared with their urban counterparts, a larger proportion of rural females suffer from insufficient iodine levels, highlighting the need for targeted interventions in rural areas to improve iodine nutrition.

The regression analysis results are presented in Table 5. The results revealed the significant impact of respondents' residence, knowledge, attitudes, and practices regarding iodine on their UIC levels. The calculated $R^2$ of 0.441 indicates that approximately 44.1% of the variability in UIC is explained by the model.

The scatter plots in Fig 4 show the relationships between the KAP scores of pregnant women with respect to iodine deficiency disorders and urinary iodine concentration, along with the correlation coefficients. The KAP scores reflect the population's general awareness and behaviors toward iodine consumption. Higher knowledge scores correlate with better

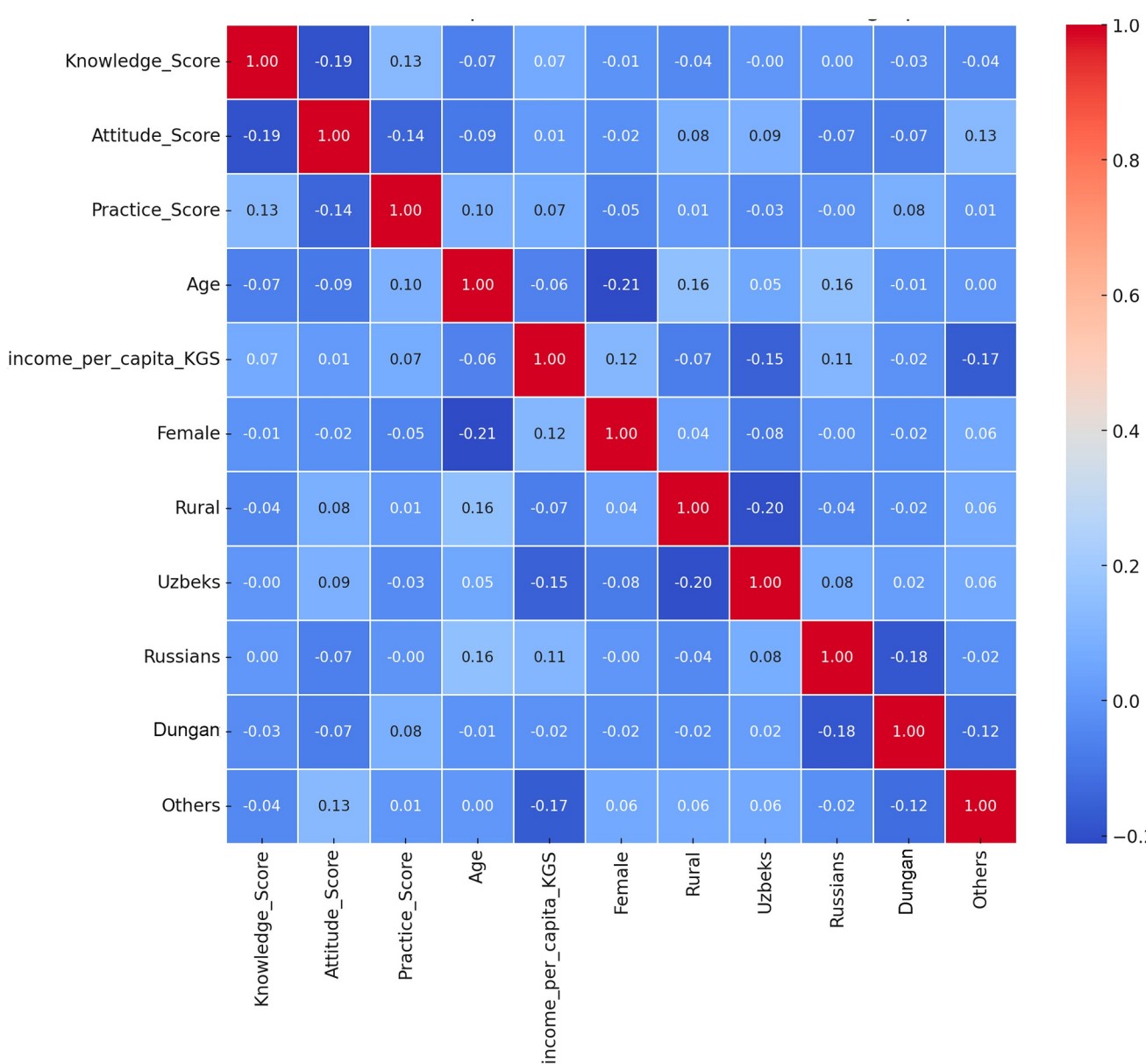

**Fig 3. The correlation matrix heatmap for KAP scores related to iodine deficiency and the sociodemographic characteristics of the respondents in Kyrgyzstan.**

**Table 4. The urinary iodine concentration (UIC) of the study population, incl. pregnant women.**

| UIC (μg/L) | Respondents | Urban (Mean±SD) | Rural (Mean±SD) | Total (Mean±SD) |
|---|---|---|---|---|
| **UIC (μg/L)** | All participants | 105.3 ± 23.5 | 88.7 ± 20.1 | 97.0 ± 22.3 |
| | Pregnant women | 203.47 ± 82.20 | 153.31 ± 90.06 | 177.75 ± 89.75 |
| **Sufficient (%)** | All participants | 68.5 ± 7.8 | 52.4 ± 6.4 | 60.5 ± 7.1 |
| | Pregnant women | 77.4 ± 9.4 | 49.3 ± 7.1 | 63.0 ± 8.0 |
| **Insufficient (%)** | All participants | 31.5 ± 7.8 | 47.6 ± 6.4 | 39.5 ± 7.1 |
| | Pregnant women | 22.6 ± 9.4 | 50.7 ± 7.1 | 37.0 ± 8.0 |

**Table 5. Associations between some sociodemographic characteristics, KAP scores associated with iodine deficiency and urinary iodine concentrations in the study population.**

| Variable | Coefficient | Std. Error | t | p | 95% CI |
|---|---|---|---|---|---|
| Intercept (const) | -10.0810 | 16.309 | -0.618 | 0.537 | [-42.104, 21.942] |
| Age | -0.8137 | 0.277 | -2.939 | 0.003 | [-1.357, -0.270] |
| Sex | -3.2133 | 4.965 | -0.647 | 0.518 | [-12.962, 6.535] |
| Residency | -26.8442 | 5.082 | -5.282 | 0.000 | [-36.822, -16.866] |
| Knowledge | 25.1352 | 5.216 | 4.819 | 0.000 | [14.893, 35.377] |
| Attitude | 37.4587 | 2.146 | 17.453 | 0.000 | [33.245, 41.673] |
| Practice | 17.9076 | 2.329 | 7.688 | 0.000 | [13.334, 22.481] |

attitudes, while practical behavior scores remain suboptimal, especially in rural areas, highlighting a gap between knowledge and action.

Although knowledge about iodine didn't strongly affect iodine levels, more positive attitudes were linked to higher iodine concentrations.

## Discussion

This study aimed to assess the knowledge, attitudes, and practices (KAP) regarding iodine deficiency among the population, with a focus on pregnant women in Kyrgyzstan, and to correlate these factors with the urinary iodine concentration (UIC). The findings highlight significant gaps in knowledge and practices related to iodine intake, despite mandatory iodization programs.

### KAP scores related to iodine deficiency

This study highlights significant disparities in iodine-related knowledge, attitudes, and practices (KAP) between urban and rural populations. While urban participants demonstrated better knowledge and attitudes toward iodine intake, practical iodine consumption behaviors, particularly in rural areas, remained inadequate despite mandatory iodization programs [14]. These scores suggest a disconnect between knowledge and behavior, particularly in rural populations. Similar findings in Portugal were reported by Pinheiro et al. (2021), who

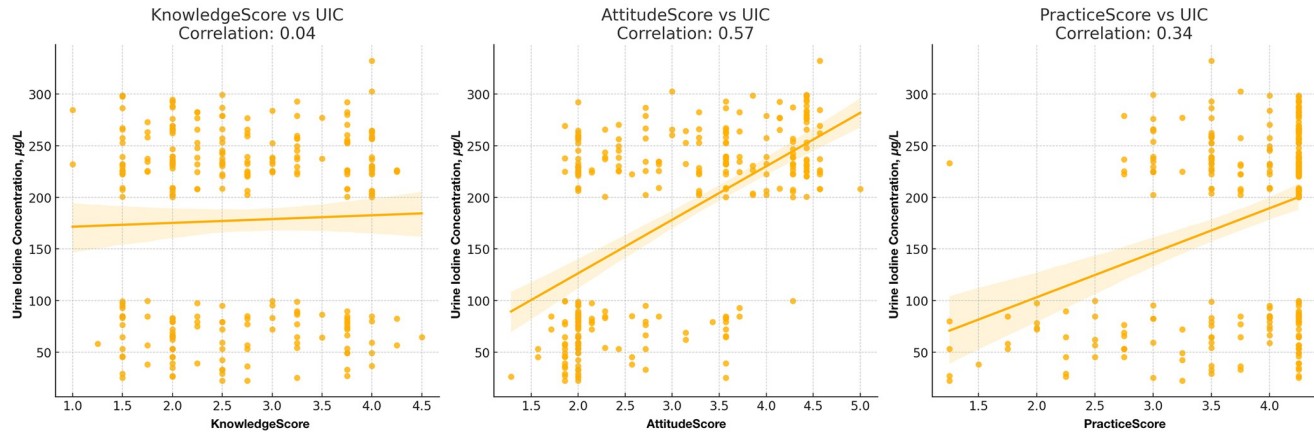

**Fig 4. Relationships between knowledge, attitudes, and practices scores of pregnant women with respect to iodine deficiency disorders and urinary iodine concentration (UIC, µg/L).**

demonstrated that although urban populations had better knowledge, rural areas still face significant practical barriers to iodine consumption [14].

Our findings align with those of previous studies indicating persistent iodine deficiency in regions with inadequate natural iodine sources [10–14]. In Bangladesh [15] and India [11, 16], the mean knowledge scores were slightly higher in urban areas than in rural areas, similar to our findings. However, overall awareness of iodine deficiency is generally low across these countries [11, 15, 16]. In India, rural practices were particularly inadequate due to limited access to iodized salt, mirroring the situation in Kyrgyzstan [16].

The moderate correlation of family income with both attitudes and practice scores suggests that socioeconomic status impacts health practices and attitudes. This finding is consistent with the findings of Mehran et al. (2012) [17]. Public health policies might need to consider economic support alongside educational efforts to improve iodine intake behaviors [17–19].

The study revealed a negative correlation between the age and practice score of the respondents, which could indicate a generational gap in practices, possibly due to changing norms or accessibility issues among older individuals. Karmakar et al. (2019) came to the same conclusions [16]. This generational gap may be addressed through targeted educational campaigns that consider the specific needs and barriers faced by different age groups.

Similar to the findings of Sherriff et al. (2020) in Australia [12], the varied correlations among different ethnic groups (in our study, Uzbeks, Russians and Dungans had relatively high KAP scores) underscore the need for culturally tailored interventions in health education and practice. These interventions should respect and incorporate cultural values and practices to be more effective.

Our study indicates that while rural pregnant females tend to have slightly better knowledge scores, urban pregnant females have more positive attitudes toward iodine-related health issues. However, the practical behaviors related to iodine consumption are consistent across both urban and rural areas. These insights can help tailor specific educational and intervention programs to improve iodine-related health behaviors among pregnant females in different regions.

Research indicates that pregnant women have greater knowledge about iodine-related health issues than does the general population does, suggesting that pregnancy might be a period in which women seek or receive more health-related information. This finding agrees with the study of Amiri et al. (2017) in Iran [18]. They demonstrate more positive attitudes toward iodine-related health issues than the general population does, likely due to increased awareness and concern for their own health and that of their unborn child.

Although pregnant women in the study had higher knowledge levels and more positive attitudes, practical barriers such as limited access to iodized salt may explain their lower iodine intake compared to the general population. This discrepancy might be due to insufficient public health interventions targeting pregnant women specifically [20]. Addressing these barriers is crucial for improving iodine intake practices.

The analysis indicated that pregnant women have better knowledge of and more positive attitudes toward iodine-related health issues than the general population does. This finding is comparable with the study of Mohapatra et al. (2001) in India [10]. However, their practical behaviors related to iodine consumption are lower. This highlights a gap between knowledge/ attitudes and actual practices among pregnant females, suggesting a need for targeted interventions to translate their knowledge and positive attitudes into effective practices.

They demonstrate more positive attitudes toward iodine-related health issues than the general population does, likely due to increased awareness and concern for their own health and that of their unborn child.

Addressing barriers to accessing iodine-rich foods and iodized salt could improve practical behaviors among pregnant females, ultimately enhancing their health outcomes and those of their babies. Gerasimov et al. (2018) reported that targeted interventions that consider the specific needs and barriers of pregnant women are effective [21].

While there is moderate knowledge and positive attitudes toward iodine intake, practical behaviors remain suboptimal, especially among pregnant women. Addressing socioeconomic, generational, and cultural barriers and providing targeted interventions are essential for translating knowledge and attitudes into improved health practices.

Multivariate regression analysis was used to understand the relationships between the dependent variable (e.g., knowledge score) and multiple independent variables (e.g., age, income, gender). In this context, the analysis provided insights into how each sociodemographic factor impacts the KAP scores.

### Urinary iodine concentration

The mean UIC was within the insufficient range for a significant portion of the study population, particularly in rural areas. Our study revealed that urban participants have a greater mean UIC than rural participants do, indicating better iodine nutrition among the urban population. A greater percentage of urban participants than rural participants have sufficient iodine levels. Moreover, a significant portion of the rural population has insufficient iodine levels, highlighting the disparity between urban and rural areas. Pregnant respondents in urban areas have a considerably higher mean UIC than those in rural areas do. Overall, pregnant women have higher UICs than the general population does, reflecting increased iodine requirements during pregnancy.

This finding indicates significant disparities in iodine nutrition between urban and rural populations, with pregnant women generally having higher UICs than the general population does. These findings are consistent with those of global and regional studies, highlighting the need for targeted interventions to address iodine deficiency, particularly in rural areas. The UIC values for pregnant women in Uzbekistan are similar to the rural UIC values in Table 4, suggesting that the rural populations in the referenced study may have similar iodine nutrition challenges [34]. The findings of the evaluation of urinary iodine levels after the implementation of wheat flour and salt fortification programs in five Central Asian countries and Mongolia are similar to the differences observed between urban and rural populations in our study [22]. The findings of the present study are consistent with the results of studies conducted in China [23–25], the UK [26], Croatia [27], Spain [28], Japan [29], Cambodia [30], and India [31, 32]. Comparisons with other regions, such as Uzbekistan and Nepal, reveal similar trends where iodine deficiency remains a concern despite mandatory iodization efforts. These findings underscore the importance of continuous monitoring and tailored interventions in biogeographically similar regions [33, 34].

### Associations between the KAP score and the urine iodine concentration

The positive correlation between the KAP score and UIC underscores the importance of enhancing public awareness and education related to iodine deficiency. The disparity between urban and rural areas indicates that targeted interventions are needed to address regional differences in iodine intake and deficiency risk.

Similarly, research conducted in Nepal has shown a strong positive correlation between attitudes and iodine sufficiency, emphasizing the need for positive health attitudes in combating deficiencies [33]. These findings parallel our study's results, which also revealed a significant correlation between positive attitudes and sufficient UIC. They indicated that despite

mandatory iodization programs, rural populations still experienced significant iodine deficiency due to poor implementation and a lack of education [33]. Studies from various countries indicate that iodine deficiency remains a concern, particularly in rural areas and among pregnant women [14–18, 20, 35].

Merely having knowledge about iodine may not translate into higher iodine intake or retention, as measured by UIC [35]. The very weak correlation between knowledge scores and UIC indicates that knowing about iodine-related health issues alone is not enough to ensure adequate iodine intake. This underscores the need for interventions that go beyond knowledge dissemination. Moreover, positive attitudes toward iodine and its importance are moderately associated with higher UIC levels. It aligns with the study of Sáez et al. (2024), which highlights the strong association between attitudes toward iodine and better iodine nutrition status, especially among pregnant women. This implies that pregnant respondents with better attitudes toward iodine are likely to have better iodine intake, reflected in their UIC levels. The moderate positive correlation between the attitude scores and UIC highlights the importance of fostering positive attitudes toward iodine consumption. Educational programs that improve attitudes toward iodine can potentially improve iodine nutrition [18].

The respondents' practical behaviors related to iodine consumption, such as using iodized salt and consuming iodine-rich foods, are moderately associated with higher UIC levels, which corresponds to the findings of Mirmiran et al. (2013) [20]. These findings suggest that good practices are linked to better iodine nutrition in pregnant women.

The correlation between KAP and iodine status aligns with global evidence emphasizing the role of public health education in improving micronutrient intake.

**Strengths and limitations.** The study utilized a robust sampling method and comprehensive data collection techniques, including structured questionnaires and urine sample analysis, ensuring a representative sample and reliable data. By focusing on pregnant women, the study addressed a particularly vulnerable group, providing valuable insights into their specific needs and challenges. The study's correlation and regression analyses provided a clear understanding of the relationship between KAP scores and iodine status, emphasizing the importance of public health education.

While the cross-sectional design limits causality, the findings provide robust insights into the current state of iodine nutrition and the factors influencing KAP scores. The reliance on self-reported data for KAP scores may introduce response bias, potentially affecting the accuracy of the findings. While the study included both urban and rural areas, the variability within these regions may not be fully captured, limiting the generalizability of the results.

**Implications for public health.** The findings of this study underscore the critical need for comprehensive public health campaigns aimed at enhancing knowledge and practices related to iodine intake. A key priority should be the implementation of targeted education programs that emphasize the importance of iodine for overall health, particularly for vulnerable groups such as pregnant women, children, and those living in rural areas. These educational initiatives should leverage various platforms, including community health workshops, school-based programs, and mass media campaigns, to reach a broad audience.

Furthermore, this study highlights the necessity of improving the accessibility and affordability of iodized salt and iodine-rich foods in rural areas, where iodine deficiency is more prevalent. Public health policies should focus on ensuring that iodized salt is available and affordable in all regions, supported by regular monitoring to assess iodine status across different populations. Strengthening existing iodization programs and enhancing regulatory frameworks to ensure compliance can significantly mitigate the risks associated with iodine deficiency.

Additionally, there is a need for regular surveillance of iodine status, particularly among pregnant women, to guide public health strategies effectively. Health care providers should be trained to educate pregnant women about the importance of iodine intake and provide guidance on dietary sources of iodine. Integrating iodine supplementation into routine antenatal care could further support adequate iodine intake during pregnancy.

## Conclusion

This study provides valuable insights into the knowledge, attitudes, and practices regarding iodine deficiency among the population of Kyrgyzstan, with a particular focus on pregnant women, and their correlation with the urinary iodine concentration. The findings reveal significant gaps in knowledge and suboptimal practices related to iodine intake, despite the presence of mandatory iodization programs. The disparity between urban and rural populations, as well as the differences among various sociodemographic groups, highlights the need for tailored public health interventions.

The correlation between KAP scores and urinary iodine concentration (UIC) suggests that improving knowledge and fostering positive attitudes toward iodine consumption can lead to better iodine nutrition. However, the weak correlation between knowledge and UIC indicates that knowledge alone is insufficient. Practical measures, such as enhancing the availability of iodized salt and addressing barriers to its consumption, are crucial for translating knowledge into effective practices.

Ultimately, this study emphasizes the importance of a multifaceted approach to combating iodine deficiency in Kyrgyzstan. Targeted public health interventions are essential to improve iodine intake and reduce the burden of iodine deficiency disorders in Kyrgyzstan. Public health authorities must prioritize education, accessibility, and regular monitoring to ensure adequate iodine intake across all segments of the population. By addressing these key areas, Kyrgyzstan can make significant strides in reducing the burden of iodine deficiency disorders and improving the overall health and well-being of its people.

Public health authorities must prioritize education, accessibility, and regular monitoring to ensure adequate iodine intake across all segments of the population. By addressing these key areas, Kyrgyzstan can make significant strides in reducing the burden of iodine deficiency disorders.

## Supporting information

**S1 Data. Sample data of pregnant women.**
(CSV)

**S2 Data. Sample data of the general population.**
(CSV)

## Author Contributions

**Conceptualization:** Yulia Chyngyshpaeva, Don Eliseo III Lucero-Prisno, Kenesh Dzhusupov.

**Data curation:** Yulia Chyngyshpaeva, Zhokhongir Dzhaliev, Zhyparkul Derbishalieva, Muiz Ibrahim.

**Formal analysis:** Yulia Chyngyshpaeva, Zhyparkul Derbishalieva, Muiz Ibrahim.

**Investigation:** Yulia Chyngyshpaeva, Zhyparkul Derbishalieva, Muiz Ibrahim.

**Methodology:** Don Eliseo III Lucero-Prisno, Kenesh Dzhusupov.

**Project administration:** Kenesh Dzhusupov.

**Resources:** Yulia Chyngyshpaeva.

**Software:** Yulia Chyngyshpaeva, Zhokhongir Dzhaliev, Muiz Ibrahim.

**Supervision:** Don Eliseo III Lucero-Prisno, Kenesh Dzhusupov.

**Validation:** Zhokhongir Dzhaliev, Muiz Ibrahim.

**Visualization:** Yulia Chyngyshpaeva, Zhokhongir Dzhaliev.

**Writing – original draft:** Yulia Chyngyshpaeva.

**Writing – review & editing:** Don Eliseo III Lucero-Prisno, Kenesh Dzhusupov.

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
