## [Decision Letter · Decision Letter 0]

20 Sep 2024

PONE-D-24-31353The state of iodine deficiency in Kyrgyzstan: insights from studies of knowledge, attitudes and practicesPLOS ONE

Dear Dr. Dzhusupov,

Thank you for submitting your manuscript to PLOS ONE. After careful consideration, we feel that it has merit but does not fully meet PLOS ONE’s publication criteria as it currently stands. Therefore, we invite you to submit a revised version of the manuscript that addresses the points raised during the review process.

We look forward to receiving your revised manuscript.

Kind regards,

Ramesh Athe, PhD

Academic Editor

PLOS ONE

Journal Requirements:

Reviewers' comments:

Reviewer's Responses to Questions

**Comments to the Author**

1. Is the manuscript technically sound, and do the data support the conclusions?

Reviewer #1: Partly

Reviewer #2: Yes

2. Has the statistical analysis been performed appropriately and rigorously? 

Reviewer #1: I Don't Know

Reviewer #2: Yes

3. Have the authors made all data underlying the findings in their manuscript fully available?

Reviewer #1: No

Reviewer #2: Yes

4. Is the manuscript presented in an intelligible fashion and written in standard English?

Reviewer #1: Yes

Reviewer #2: Yes

5. Review Comments to the Author

Reviewer #1: The MS ID # PONE-D-24-31353 titled “The state of iodine deficiency in Kyrgyzstan: insights from studies of knowledge, attitudes and practices” by Kenesh Dzhusupov et al., identified significant knowledge gap related to iodine nutrition and in the Kyrgyzstan population.

Although the results are encouraging,

The following minor comments could be addressed before the manuscript acceptance for publication.

1. The figures in poor quality and hard to review.

2. Not clear whether the thyroid hormone levels and other health issues associated with UIC in these subjects?

3. Not sure whether the IDD in sub clinical or clinical level?

4. Is there any relationship between of low iodine and economic status?

Reviewer #2: Thanks for the opportunity to review the manuscript titled as “The state of iodine deficiency in Kyrgyzstan: insights from studies of knowledge, attitudes and practices (PONE-D-24-31353). The paper is quiet interesting and of intriguing interest in the maternal and child healthcare issues and concern. Though the paper is quiet relevant in the contemporary times, I have certain concerns and queries which needs to be addressed by the authors. The comments are listed in detail as under:

1. The introduction effectively sets the stage for the study. However, it could be improved by discussing the potential impact of cultural factors on iodine deficiency and how this study addresses those factors.

2. The literature cited is comprehensive, but more recent studies on iodine deficiency and interventions could be included. This would help contextualize the findings within the broader global effort to combat iodine deficiency.

3. The methodology is robust, but more details about the training of data collectors and the validation process of the questionnaire would enhance the reliability of the data. Additionally, the rationale for the chosen statistical methods should be explained in more detail.

4. The sampling method is well described, but it might be helpful to clarify how the pilot study influenced the final design of the questionnaire. Were there any significant changes based on the pilot study results?

5. The multivariate regression model used to assess predictors of urinary iodine concentration (UIC) includes several variables (residence, knowledge, attitudes, practices, etc.). However, there is no mention of how multicollinearity between these predictors was assessed. Multicollinearity can inflate standard errors and make it difficult to determine the significance of individual predictors.

6. The study uses knowledge, attitude, and practice (KAP) scores as continuous variables in the regression analysis. While this is appropriate, the paper does not discuss whether these scores meet the assumptions of normality required for regression analysis. Additionally, the urinary iodine concentration (UIC) is also treated as a continuous variable, and its distribution is not examined.

7. The presentation of the results is clear, but some sections could benefit from more detailed explanations. For example, in Table 3, it would be helpful to briefly explain the significance of the KAP scores and how they relate to each other in the text.

8. Consider improving the readability of figures and tables. For example, Fig 4 (Relationships between KAP scores and UIC) could be more informative if the axes were labelled more clearly and the relationship explained in more detail within the figure caption.

9. The discussion could be more concise. While it effectively interprets the findings, it tends to repeat points already made in the results section. Focus on the most critical findings and their implications.

10. The discussion effectively highlights the gaps in knowledge and practices. However, it would be beneficial to compare the findings with other similar regions to give a more global perspective. Also, discussing the limitations in greater detail, particularly the cross-sectional design and potential biases, would strengthen the paper.

11. Though the paper is structured properly, and the objectives are clear. However, the discussion section could benefit from more concise and focused arguments. Streamlining the findings and their implications will help the reader better understand the key takeaways.

12. The conclusions are sound but could be more actionable. It would be helpful to provide specific recommendations for policy changes or interventions based on the study’s findings. The conclusion should include more specific recommendations for public health interventions. Additionally, emphasizing the importance of ongoing monitoring and the role of healthcare providers in education could strengthen this section.

13. While the language is generally clear, there are a few areas where sentences could be simplified for better readability. Proofreading for minor grammatical errors is recommended.

14. Ensure that all references are up-to-date and relevant to the study. Some references could be replaced with more recent studies if available.

The paper provides valuable insights into iodine deficiency in Kyrgyzstan, but it could be enhanced by clarifying the discussion, offering more actionable conclusions, and ensuring all methods and data are transparently presented. Overall, it is a well-executed study that could significantly contribute to public health interventions in the region.

6. PLOS authors have the option to publish the peer review history of their article (what does this mean?). If published, this will include your full peer review and any attached files.

Reviewer #1: No

Reviewer #2: **Yes**

---

## [Author Response · Author response to Decision Letter 0]

22 Oct 2024

Reviewer #1:

Comment 1.1: The figures in poor quality and hard to review. 

Response: All figures are generated by a software and we cannot improve their quality except making bold the letters in names of axes. 

Comment 1.2: Not clear whether the thyroid hormone levels and other health issues associated with UIC in these subjects?

Response: In our study, we did not check thyroid hormone levels and other health issues

Comment 1.3: Not sure whether the IDD in subclinical or clinical level?

Response: In the questionnaire, respondents were asked to answer, were they ever diagnosed any IDD.

Comment 1.4: Is there any relationship between low iodine and economic status?

Response: This issue is not highlighted in this paper since it is not the task of this paper. However, we did explore the association between UIC level and economic status (family income per capita) and highlighted it in our second manuscript submitted to this journal following this one.

Reviewer #2:

Comment 2.1: The introduction effectively sets the stage for the study. However, it could be improved by discussing the potential impact of cultural factors on iodine deficiency and how this study addresses those factors.

Response: We underlined the role of cultural factors in determining IDDs – lines 72-76.

Comment 2.2: The literature cited is comprehensive, but more recent studies on iodine deficiency and interventions could be included. This would help contextualize the findings within the broader global effort to combat iodine deficiency.

Response: Recent studies on iodine deficiency and interventions were mentioned in the review with focus on the findings within the broader global effort to combat iodine deficiency - lines 59-63.

Comment 2.3: The methodology is robust, but more details about the training of data collectors and the validation process of the questionnaire would enhance the reliability of the data. Additionally, the rationale for the chosen statistical methods should be explained in more detail.

Response: We improved description of the training of data collectors and the validation process of the questionnaire – lines 142-150.

Comment 2.4: The sampling method is well described, but it might be helpful to clarify how the pilot study influenced the final design of the questionnaire. Were there any significant changes based on the pilot study results?

Response: We clarify how the pilot study influenced the final design of the questionnaire and that it relusted in rephrasing and adjustment of its language – lines 141-145.

Comment 2.5: The multivariate regression model used to assess predictors of urinary iodine concentration (UIC) includes several variables (residence, knowledge, attitudes, practices, etc.). However, there is no mention of how multicollinearity between these predictors was assessed. Multicollinearity can inflate standard errors and make it difficult to determine the significance of individual predictors.

Response: We described how multicollinearity between these predictors was assessed in lines 171-174.

Comment 2.6: The study uses knowledge, attitude, and practice (KAP) scores as continuous variables in the regression analysis. While this is appropriate, the paper does not discuss whether these scores meet the assumptions of normality required for regression analysis. Additionally, the urinary iodine concentration (UIC) is also treated as a continuous variable, and its distribution is not examined.

Response: We described testing the assumptions of normality required for regression analysis in lines 184-186.

Comment 2.7: The presentation of the results is clear, but some sections could benefit from more detailed explanations. For example, in Table 3, it would be helpful to briefly explain the significance of the KAP scores and how they relate to each other in the text.

Response: We gave detailes of the significance of the KAP scores and how they relate to each other in lines 213-216.

Comment 2.8: Consider improving the readability of figures and tables. For example, Fig 4 (Relationships between KAP scores and UIC) could be more informative if the axes were labelled more clearly and the relationship explained in more detail within the figure caption.

Response: We have labelled axes S4 Fig clearer to indicate the relationship between KAP scores and UIC and added sentences – lines 301-306.

Comment 2.9: The discussion could be more concise. While it effectively interprets the findings, it tends to repeat points already made in the results section. Focus on the most critical findings and their implications.

Response: To avoid repeated points, we deleted some lines in the Results (lines 311-316) and in the Discussion sections (lines 330-335, 367-370, 435-437, 461-462) and reformulate some of them (lines 323-326, 370-374), added some sentences – lines 326-333, 462-464.

Comment 2.10: The discussion effectively highlights the gaps in knowledge and practices. However, it would be beneficial to compare the findings with other similar regions to give a more global perspective. Also, discussing the limitations in greater detail, particularly the cross-sectional design and potential biases, would strengthen the paper.

Response: Despite that in our manuscript, we have already compared our findings with other similar regions (lines 336-341, 343-345, 351-354, 363-366, 377-379, 408-416), we wrote additional sentences (lines 328-330, 417-420, 442-444).

Comment 2.11: Though the paper is structured properly, and the objectives are clear. However, the discussion section could benefit from more concise and focused arguments. Streamlining the findings and their implications will help the reader better understand the key takeaways.

Response: The study findings are streamlined in the introduction of the Discussion section and in the beginning each subsections (lines 323-324, 326-330, 356-359, 361-363, 398-407, 423-426). We reformulate some of them – lines 323-330. Their implications are give in lines 468-485.

Comment 2.12: The conclusions are sound but could be more actionable. It would be helpful to provide specific recommendations for policy changes or interventions based on the study’s findings. The conclusion should include more specific recommendations for public health interventions. Additionally, emphasizing the importance of ongoing monitoring and the role of healthcare providers in education could strengthen this section.

Response: We added two sentences with specific recommendations for policy changes or interventions providers in education based on the study’s findings stressing their improtance in lines 507-510.

Comment 2.13: While the language is generally clear, there are a few areas where sentences could be simplified for better readability. Proofreading for minor grammatical errors is recommended.

Response: For better readability and decrease redundancy in the text we deleted some sentences and added new ones – lines 310-316, 323-330.

Comment 2.14: Ensure that all references are up-to-date and relevant to the study. Some references could be replaced with more recent studies if available.

Response: We updated 11th, 25th, 28th, 29th, 30ties and 31st references and added anew, 26th and 33rd references.

---

## [Editor Report · Decision Letter 1]

1 Nov 2024

The state of iodine deficiency in Kyrgyzstan: insights from studies of knowledge, attitudes and practices

PONE-D-24-31353R1

Dear Dr. Dzhusupov,

We’re pleased to inform you that your manuscript has been judged scientifically suitable for publication and will be formally accepted for publication once it meets all outstanding technical requirements.

Kind regards,

Ramesh Athe, PhD

Academic Editor

PLOS ONE
---

## [Editor Report · Acceptance letter]

9 Nov 2024

PONE-D-24-31353R1 

PLOS ONE

Dear Dr. Dzhusupov, 

I'm pleased to inform you that your manuscript has been deemed suitable for publication in PLOS ONE. Congratulations! Your manuscript is now being handed over to our production team.

Kind regards, 

on behalf of

Dr. Ramesh Athe 

Academic Editor

PLOS ONE